

# Thermography analysis as a tool for assessing thermal asymmetries and temperature changes after therapy in patients with stroke: a pilot study

Luis Augusto Silva Zendron[1], Marta Gómez Mateos[2], Beatriz María Bermejo Gil[2], Andrea Calleja Caballero[2], Vanesa Santos Rodríguez[2], Fátima Pérez-Robledo[2] and Ana María Martín Nogueras[2]

[1] Department of Computer Science, Universidad de Salamanca, Salamanca, Castilla y León, Spain
[2] Department of Nursery and Physiotherapy, Faculty of Nursing and Physiotherapy, Universidad de Salamanca, Salamanca, Spain

## ABSTRACT

This pilot quasi-experimental study investigates the potential of infrared thermography as a non-invasive tool for assessing thermal asymmetries in patients with hemiplegia following stroke. Ten participants underwent thermographic imaging using a FLIR C5 camera before and after a lower-limb muscle-strength intervention. Thermal data were processed and analyzed with ThermImageJ software, following the TISEM protocol to ensure the precision of temperature measurements within predefined regions of interest (ROI). The primary aim was to evaluate whether thermography could reliably detect thermal discrepancies between the affected and unaffected sides of the body and whether these differences respond to targeted physical therapy. The results demonstrated significant baseline asymmetries between both sides of the body, which were notably reduced after the strength intervention. These findings suggest that muscle-strength training may contribute to improved thermal symmetry and that thermography is sensitive enough to detect such changes. While the outcomes are promising, larger-scale studies with extended follow-up are necessary to confirm these preliminary findings. Nonetheless, infrared thermography is an effective complementary method for monitoring physiological responses to rehabilitation in stroke patients.

## INTRODUCTION

Acquired brain injury (ABI) is one of the most common disabling conditions worldwide, and it is a leading cause of death from non-communicable diseases (*GBD 2019 Stroke Collaborators, 2021*). In Europe, it is estimated that more than two million people are affected by this condition. The most common etiology is stroke, followed by traumatic brain injury (*Wafa et al., 2020*). Stroke encompasses a range of consequences resulting from brain lesions that disrupt normal functioning in the affected areas (*Rubio & Atarés, 2019*). After a stroke, a series of neuro-physiological and structural changes occur, leading to a

Corresponding authors
Fátima Pérez-Robledo,
fatima_pr@usal.es
Luis Augusto Silva Zendron,
luisaugustos@usal.es

diverse number of physical and cognitive complications (*Joubran, Bar-Haim & Shmuelof, 2022*).

Among the most prevalent sequelae of stroke is hemiplegia (*Sun et al., 2021*), in which the patient experiences full or partial paralysis of one part of the body with increased motor and sensory impairment. Signs of hemiplegia include flaccidity, which mainly appears in acute phases, spasticity (a speed-dependent increase in muscle tone), motor or sensory paresis, and deficits in coordination and balance control. This condition compromises mobility, sensitivity, nerve transmission, and blood circulation (*Francisco et al., 2021*).

An underexplored but clinically relevant manifestation of this dysregulation is the asymmetries between the affected and unaffected sides of the body. Numerous studies indicate a significant thermal asymmetry between the plegic side and the contralateral side (*Alfieri et al., 2017*). These thermal differences affect patients' thermal sensation and may influence their ability to feel and move. Proper blood circulation is essential for ensuring suitable tissue function by supplying the necessary oxygen and nutrients. An alteration in this system can cause different body dysfunctions in one side of the body, thus preventing correct movement patterns (*da Silva Dias et al., 2021*).

One of the techniques used to assess these asymmetries is thermographic imaging, which accurately analyzes the thermal differences between various body segments (*Hegedűs, 2018*).

Infrared thermography is a non-invasive, non-contact imaging technique that detects and measures infrared radiation emitted by the surface of objects or the human body. It converts temperature variations into visual images known as thermograms, which display heat distribution by assigning specific colors to different temperature levels. Because any object with a temperature above absolute zero emits infrared radiation, thermography enables physiological assessment without requiring physical contact or exposing the subject to ionizing radiation. This technique can detect subtle differences in skin surface temperature, particularly thermal asymmetries greater than 0.5 °C, which may indicate abnormal physiological or pathological conditions. Consequently, thermography can identify localized areas of hyperthermia and hypothermia (*Usamentiaga et al., 2014*; *Sánchez-Sánchez et al., 2019*).

Although thermography in healthcare is not yet widely explored, several studies support its efficacy as a diagnostic tool for vascular disorders, breast cancer screening, complex regional pain syndrome, and diabetic neuropathy (*Lahiri et al., 2012*). Recent literature supports the application of infrared thermography in assessing and monitoring stroke-related sequelae (*Podlasek et al., 2025*). The applications of thermography in physical therapy have also been studied, assessing aspects such as the relationship between thermography and motor and sensory sequelae in patients with brain injury (*Cabizosu et al., 2024a*; *Cabizosu et al., 2024b*). However, there is still limited scientific literature on this subject, and systematic reviews suggest further research in the physical therapy area (*Grotto, Luceño-Sánchez & Cabizosu, 2023*).

The sensation of coldness in the limb affected by hemiparesis is a common symptom among stroke survivors. According to some sources, it occurs in 64% of cases and can be between 1 and 5 degrees lower than the temperature of the unaffected limb (*Alfieri et al., 2017*). Infrared thermography can monitor these temperature differences, capturing

the differences between the affected and contralateral limbs. Additionally, it can assist in identifying localized inflammation or circulatory problems, offering valuable insights into the patient's vascular and physiological status.

Physical therapy is an approach aimed at reducing or modifying the after-effects of stroke. Through physical agents and movement, the aim is to modify the functional state of the patient (*Fang et al., 2003*), generating greater body symmetry enables more precise movements and improved mobility recovery for the patient post-injury. In this field, various physiotherapeutic techniques have been proven effective in reducing this asymmetry, such as hydrotherapy (*Zhu et al., 2016*), virtual reality (*Peláez-Vélez et al., 2023*), or muscle training (*Lattouf et al., 2021*). All these techniques have proven effective in improving different movement and posture patterns, such as gait and balance.

One of the most scientifically supported techniques is muscle strength training. This practice has been proven to effectively improve the functionality and quality of life for people who have had a stroke. It can reduce the energy needed to perform daily activities (*Kim et al., 2019*). Muscle strength is effective in gaining muscle mass, essential when brain damage occurs, and improves balance and walking ability. It benefits walking speed, increasing endurance, and covering longer distances (*Han et al., 2017*; *Bale & Strand, 2008*; *Hunnicutt et al., 2016*).

However, the potential of strength training on thermal asymmetry in these patients has not yet been systematically explored. This training may promote better symmetry between the two sides of the body, potentially improving blood circulation and explaining the observed functional improvements. Our study aims to contribute to the emerging scientific literature by examining whether infrared thermography can detect changes before and after a physical therapy session.

In addition, this study explores the potential role of deep learning techniques for thermographic image analysis. Machine learning algorithms have shown great potential for segmenting and detecting differences in medical images (*Chen et al., 2022*; *Al Husaini, Habaebi & Islam, 2024*). We hypothesize that integrating these techniques in the analysis of thermograms will allow better detection of thermal asymmetries in stroke patients.

This study investigates the utility of infrared thermography in identifying temperature asymmetries in individuals with hemiplegia following a stroke. Specifically, it examines whether a single session of lower-limb strength training can induce measurable thermal changes between the two sides of the body. In parallel, the study explores the feasibility of incorporating deep learning techniques to enhance the analysis of thermographic data.

## METHODOLOGY

This quasi-experimental pilot study aimed to analyze baseline and immediate thermal responses to a muscle strength training session in individuals with hemiplegia.

### Participants

The research team recruited a convenience sample of 10 individuals with chronic stroke from a local association for acquired brain injury, all of whom were engaged in regular physiotherapy. Inclusion criteria required participants to be in the chronic stage of stroke

recovery (more than one year post-event) and to exhibit motor or sensory impairments predominantly affecting one side of the body. Exclusion criteria included the inability to stand independently and the presence of inflammatory or infectious conditions at the time of assessment. Before inclusion, all participants received detailed information about the study and provided written informed consent. The Bioethics Committee of the University of Salamanca approved the study protocol (Ref: 583).

## Procedure

Researchers performed thermographic imaging using a FLIR C5 infrared camera, which offers a thermal resolution of $160 \times 120$ pixels, a frame rate of 8.7 Hz, and a spectral range of 8–14 µm. The device operates within a temperature range of $-20$ °C to 400 °C, provides an accuracy of $\pm 3\%$, and has a thermal sensitivity (NETD: Noise Equivalent Temperature Difference) below 70mK, making it well-suited for clinical thermographic evaluations (*Alfieri et al., 2023*).

Recordings took place in a closed room with the windows shut to eliminate external light or heat sources. During all sessions, they continuously monitored the room's temperature and humidity. The average temperature was $24 \pm 0.3$ °C, with a maximum of 24.4 °C and a minimum of 23 °C. The average recorded humidity was $50.5 \pm 6.4\%$, ranging from 39% to 58%. Despite efforts to maintain stability, humidity levels exhibited greater variability than temperature.

All sessions occurred in the morning, between 9:00 am and 1:00 pm. To capture thermographic images of patients standing on a carpet, the research team positioned the camera on a tripod two m away from the participant and adjusted it to a height of one m. They placed each patient 0.4 m from the wall. For imaging the soles of the feet with the patient lying down, they set the tripod 1.5 m from the participant and maintained the camera height at one m. In comparison, the patient lay on a litter positioned 0.5 m above the floor.

Researchers also collected anthropometric and socio-demographic data, including sex, age, weight, height, and body mass index (BMI), to characterize the study sample.

Before image acquisition, the researchers instructed participants to sit quietly for 15 min in the procedure room for thermal acclimatization. They asked participants to wear swimsuits or shorts to ensure full exposure of the thighs, legs, and feet. To enhance the accuracy of thermographic measurements, participants were advised to avoid physical exercise, hot showers or baths, and the application of lotions, powders, cosmetics, or other substances on their lower extremities on the day of data collection.

The research team considered additional factors that could influence thermographic recordings. They instructed participants to avoid consuming stimulants such as caffeine, alcohol, or nasal decongestants for at least two hours before the session. They also requested participants to fast during that period, except for drinking water, to minimize the risk of dehydration. This is particularly relevant since data collection occurred during the summer when elevated temperatures could increase the likelihood of heat stress. Moreover, the team advised participants not to smoke in the hours preceding the session. It encouraged

them to report any symptoms of fever or infection, as these conditions could alter body temperature and warrant exclusion from the study.

To ensure thermal equilibration, the assessor instructed participants to remain seated in a static position for 15 min before image acquisition. During this time, the assessor reminded them not to touch or scratch their lower limbs. Participants stayed barefoot and wore swimsuits to eliminate clothing as a confounding factor in the temperature readings. They also rested their feet on a towel to avoid contact with the potentially cooler floor, which might have influenced plantar temperature measurements.

Following the 15-minute acclimatization period, the assessor guided participants to stand at a marked position on a carpet to standardize body placement. The team first captured thermographic images of the anterior lower limbs and then instructed participants to reposition for posterior imaging. Finally, they asked participants to lie down to photograph the soles of their feet. The entire procedure is illustrated in Fig. 1.

Our imaging protocol adhered to the guidelines outlined in the Thermographic Imaging in Sports and Exercise Medicine (TISEM) consensus statement (*Moreira et al., 2017*), ensuring methodological rigor in image acquisition and analysis. The study followed key TISEM recommendations, including environmental control, a standardized acclimatization period within the assessment room, continuous ambient temperature and relative humidity monitoring, appropriate clothing to expose regions of interest (ROIs), and a fixed camera-to-subject distance. It is important to note, however, that one participant required postural support while standing due to hemiplegia. Table 1 summarizes the alignment between the TISEM protocol and the procedures implemented in this study.

In the images, we considered regions of interest (ROI) previously described in the scientific literature (*de Freitas Zanona et al., 2018*; *Park et al., 2022*). On the front side of the lower limb, temperature readings were taken from two points: the thigh and the leg of both limbs. On the back side, the ROIs were both the' upper and lower areas of both legs. The temperature of the most lateral area of the line cutting the anterior and posterior half of the foot (External foot) and the most medial point of this line was collected (Internal foot). To determine if there was a significant temperature difference between the two sides, a temperature gap of 0.5 °C was considered, as in other related articles (*Alfieri & Battistella, 2018*; *Alfieri et al., 2019*).

After the initial assessment, the intervention involved a 30-minute session of muscle strength training in the same room. The exercises focused on strengthening the muscles in the lower limbs, including squats, leg separations, ankle flexion and extension, and glute bridges. We used weights and elastic bands for resistance, and the physiotherapist provided the necessary resistance for those unable to use these materials.

The exercise session was individualized and adapted to each participant's functional capacity. Most participants completed the proposed exercises in sets of 10 repetitions, with a maximum of three sets per exercise when tolerated. The physiotherapist increased the exercise intensity when appropriate by adding external weight or resistance. The physiotherapist also implemented a standard 30-second rest period between sets for adequate recovery.

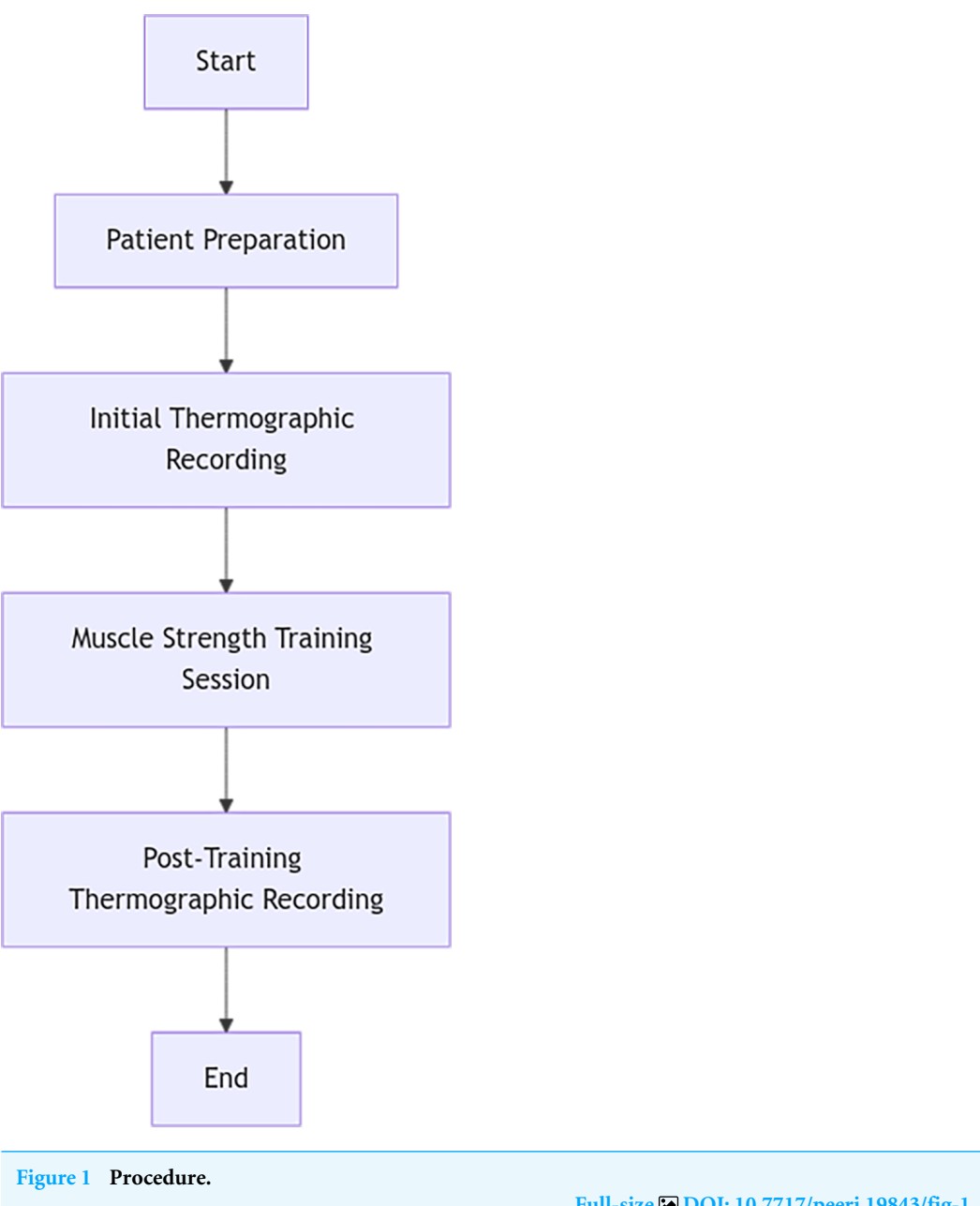

**Figure 1  Procedure.**

By the end of the session, patients had worked the main muscle groups of the lower extremity: hip flexors, extensors, abductors, and adductors; knee flexors and extensors; ankle flexors and extensors; and core musculature involving the trunk, pelvis, and hip.

After the strength training session, thermographic images were collected again using the same procedure as the initial assessment.

The analysis of thermographic images utilized FIJI software (*Schindelin et al., 2012*) with the ThermImageJ plugin, applying the standardized procedure recommended for biomedical infrared imaging to ensure accuracy and reproducibility. This protocol ensured spatial, rather than pointwise, assessment of skin temperature across defined anatomical

**Table 1** Comparison between the TISEM protocol and procedures used in this study.

| TISEM Protocol | Implementation in this study |
|---|---|
| Minimum acclimatization time of 15 min | A 15-minute seated rest period was conducted in the acquisition room. |
| Stable ambient temperature (21–24 °C) | Average recorded temperature: 24 ± 0.3 °C. |
| Monitoring of relative humidity | Recorded humidity: 50.5 ± 6.4%, with documented variations. |
| Avoid exercise, caffeine, hot showers, or smoking prior to the exam | Written instructions were provided to participants to avoid interfering factors. |
| Use of minimal clothing to expose areas of interest | Participants wore swimsuits or shorts to expose thighs, legs, and feet. |
| Camera mounted on tripod with fixed distance and height | FLIR C5 camera mounted on a tripod with standardized distances and heights according to view (frontal, posterior, plantar). |
| Camera positioned perpendicular to the ROI | FLIR C5 camera was positioned perpendicular to the ROI. |
| Time of day for image acquisition is reported | All sessions took place in the morning, from 9:00 am to 1:00 pm. |
| ROI analysis based on standardized anatomical references | ROIs defined according to previous literature (*Alfieri et al., 2019*; *De Freitas Zanona et al., 2018*; *Park et al., 2022*). |
| Use of camera validated for clinical diagnosis | A FLIR C5, comparable in features to the FLIR T650sc, was utilized (*Alfieri et al., 2023*). |
| Software is clearly described | FIJI software (*Schindelin et al., 2012*) was used in combination with the ThermImageJ plugin. |
| Extrinsic factors affecting skin temperature were described | Measurements were conducted in a closed room without external light or power; participants stood on a carpet, were instructed to fast (except water), and avoid thermal or pharmacological stimulants. |

regions, in line with established methodological practices (*Moreira et al., 2017*; *Alfieri et al., 2023*).

All thermograms acquired with the FLIR C5 camera were exported in native JPEG format and converted into calibrated thermal TIFF images using the ThermImageJ plugin, allowing for quantitative radiometric temperature mapping. Calibration parameters included an emissivity value of 0.98 for human skin and the ambient temperature and humidity recorded at acquisition time. These parameters were manually input to ensure the physical accuracy of thermal readings.

We used FIJI's ROI Manager tool to select and save the anatomical regions of interest (ROIs), ensuring consistency across image analyses. Using the software's measurement function, we calculated the average temperature for each ROI and recorded the values in a results table. We then compiled all temperature data into a spreadsheet to perform statistical analysis, where we computed temperature differences between pre- and post-training measurements. In previous studies, we defined a significant temperature change as a difference equal to or greater than 0.5 °C. We applied the appropriate statistical tests to determine the significance of the observed variations.
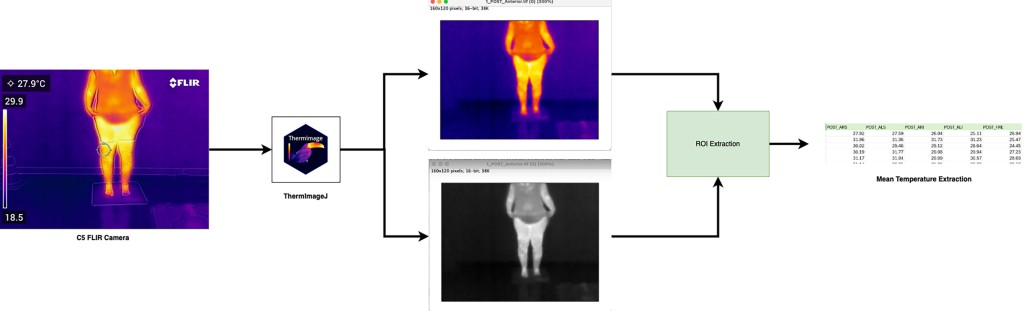

**Figure 2   Processing FLIR image to thermal using ImageJ.**

To prepare the thermographic images for analysis, we exported the native infrared images from the FLIR C5 camera to a studio computer equipped with ImageJ and the necessary plugin packages (*Tattersall, 2019*). We imported all FLIR images in their original .jpg format and converted them to calibrated thermal .tiff files using ThermImageJ. Figure 2 illustrates the image processing workflow. This protocol allowed for multidimensional thermal analysis, avoiding oversimplified point-based measurements and promoting methodological transparency.

## Statistical analysis

Statistical analysis of the results was performed using IBM SPSS Statistics (*IBM Corp., 2021*), version 28. First, a descriptive analysis of the anthropometric and sociodemographic results was performed. Mean values and their standard deviation were recorded for quantitative variables, while count and percentage were recorded for qualitative variables. The central values and dispersion obtained in infrared thermography were also described. In addition, thermal difference variables between both sides of the body were calculated. Due to the small sample size in our pilot study, nonparametric tests were used for inferential data analysis. Dependent measures analysis tests were performed to establish the relationship between pre- and post-intervention values (Wilcoxon test). In addition, Spearman's rank correlation coefficient between variables was calculated to determine possible relationships in the study. In all analyses, a 95% confidence level was used, and differences were considered significant when $p < 0.05$.

## RESULTS

In the study sample, 55% were women. The average age was 65.8 ± 9.6 years, ranging from 54 to 80 years. The average weight in the sample was 65.8 ± 9.6 kg, and the average height was 163.6 ± 10.4 cm. The average BMI of the sample was 27.2 ± 4.6 kg/m2. When analyzed according to the values established by the World Health Organization (*World Health Organization, 2025*), one participant was underweight, three were of normal weight, two were overweight, and five were within the obese range (Table 2). Half of the patients had left-sided hemiplegia, and the rest had right-sided hemiplegia.

| Table 2 Characteristics of women and men participants. | | |
| --- | --- | --- |
| | Women ($n = 6$) | Men ($n = 4$) |
| Age (years) | 64.2 ± 9.4 | 64.8 ± 10.1 |
| Height (cm) | 156.2 ± 6.1 | 169.3 ± 1.0 |
| Weight (kg) | 66.3 ± 15.0 | 76.3 ± 11.6 |
| Body Mass Index (kg/m$^2$) | 27.2 ± 5.7 | 26.6 ± 3.9 |

## Image processing

In total, 60 thermal images were analysed, six images per patient (three PRE and three POST). The 60 thermal images were converted to the .tiff file format containing only the temperature values per pixel and exported using ImageJ for customised processing Fig. 2. Region of interest (ROI) windows were created and exported for all images, with all final ROI data.

## Pre and post intervention thermographic

When analysing the thermal values recorded, it was found that the mean temperature found on the plegic side was lower than the less affected side. In all cases, this temperature was lower by about 1 °C ($p < 0.05$). This asymmetry can be seen with the naked eye in the thermal images (Figs. 3 and 4).

Significant differences were found in all ROI assessed, except in the upper anterior and lower posterior regions before intervention. However, the mean difference between both sides of the body is more significant than 0.5 °C on the lower posterior side. No statistically significant differences were found on the external side of the foot after treatment either, approximating a more excellent thermal symmetry between both body sides (Table 3).

The study examined temperature differences between the left and right sides. The most significant disparities were found on the back and front of the lower leg, with the outer side of the foot showing the most significant variation. However, implementing muscle-strength exercises effectively reduced the differences in temperature in all assessed leg regions. Even the inner side of the foot experienced a decrease to just 0.02 °C after the intervention. While the observed changes did not reach statistical significance, all participants achieved results closer to thermal symmetry (Table 4).

In analyzing the differences between both sides of the body, we found close correlations between some of them. Still, the variable of the anterior part of the thigh did not show a significant correlation with any of the others.

Before and after the intervention, differences in the internal and external regions of the foot showed a significant and positive correlation with all the other variables except for the thigh. However, the external aspect of the foot did not show a significant correlation with the posterior upper leg after intervention.

No significant correlations were observed between the lower regions of the anterior and posterior aspects of the leg, either before or after treatment. However, the anterior lower leg showed a significant and positive correlation before and after the procedure (Table 5).

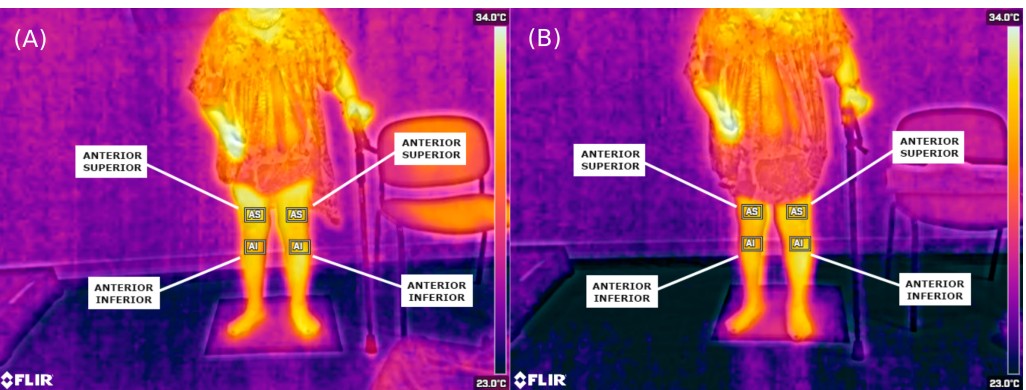

**Figure 3** **Thermographic images showing anterior lower-limb temperature distribution in a stroke patient with right hemiplegia.** (A) Pre-intervention (anterior view); (B) post-intervention (anterior view). Regions of interest (ROIs) include: AS, Anterior Superior; AI, Anterior Inferior.

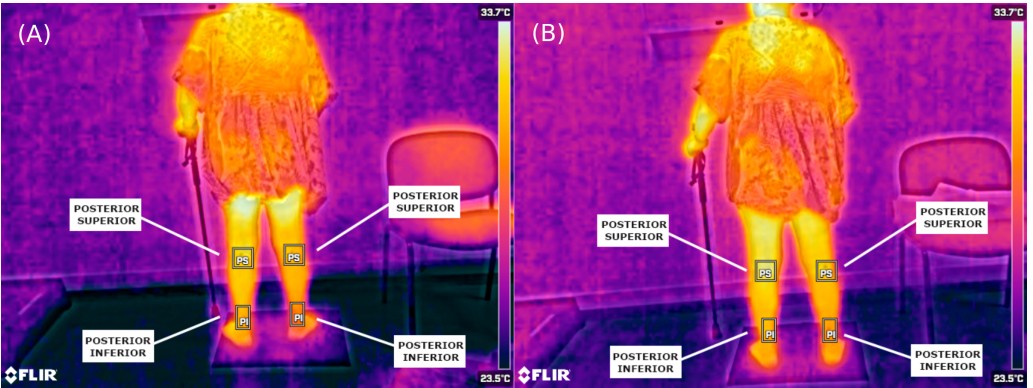

**Figure 4** **Thermographic images showing posterior lower-limb temperature distribution in the same patient.** (A) Pre-intervention (posterior view); (B) post-intervention (posterior view). ROIs include: PS, Posterior Superior; PI, Posterior Inferior.

## DISCUSSION

Thermography is a promising tool for diagnosing motor impairment in stroke patients. This non-invasive method measures body surface temperature variations and offers advantages in identifying and monitoring post-stroke motor complications.

Firstly, thermography has been proven effective in detecting thermal asymmetries that can be linked to areas affected by stroke (*Nowak, Mraz & Mraz, 2019*; *Podlasek et al., 2025*). This finding is significant because early detection of these changes enables an earlier and more personalized intervention.

In addition, thermography has several practical advantages over other diagnostic methods. Its noninvasive nature and the ability to make repeated measurements without risk to the patient make it suitable for clinical and research settings. Our results are consistent with other studies (*Sánchez-Sánchez et al., 2019*) and demonstrate the utility of

**Table 3  Pre- and post- intervention thermographic values.**

| | $n = 10$ | | | |
|---|---|---|---|---|
| | Affect | Non affect | Difference | *p* value |
| Pre-Anterior superior | 30.37 ± 1.51 | 30.69 ± 2.07 | −1.071 | 0.284 |
| Post-Anterior superior | 30.23 ± 1.17 | 31.17 ± 1.35 | −2.803 | **0.005**[**] |
| Pre-Anterior inferior | 29.30 ± 1.95 | 30.50 ± 1.87 | −2.293 | **0.022**[*] |
| Post-Anterior inferior | 29.01 ± 1.60 | 29.95 ± 1.68 | −2.395 | **0.017**[*] |
| Pre-Posterior superior | 30.54 ± 1.02 | 31.38 ± 1.30 | −2.499 | **0.012**[*] |
| Post-Posterior superior | 30.26 ± 0.92 | 31.00 ± 1.50 | −2.499 | **0.012**[*] |
| Pre-Posterior inferior | 29.18 ± 1.89 | 29.96 ± 1.77 | −1.784 | 0.074 |
| Post-Posterior inferior | 28.40 ± 1.29 | 29.18 ± 1.41 | −2.601 | **0.009**[**] |
| Pre-Internal foot | 29.67 ± 2.33 | 30.83 ± 1.93 | −2.191 | **0.028**[*] |
| Post-Internal foot | 27.30 ± 1.70 | 28.85 ± 1.76 | −2.295 | **0.022**[*] |
| Pre-External foot | 26.93 ± 1.75 | 28.15 ± 1.74 | −2.803 | **0.005**[**] |
| Post-External foot | 26.50 ± 2.15 | 26.93 ± 1.81 | −1.698 | 0.092 |

Notes.
Values pre and post presented in Mean ± Standard Deviation. Difference calculated with the Wilcoxon test.
[*]$p < 0.05$.
[**]$p < 0.01$.
Values in bold indicate statistically significant results.

**Table 4  Thermal differences between both sides of the body.**

| | $n = 10$ | |
|---|---|---|
| | Pre | Post |
| Anterior superior (AS) | 0.21 ± 1.00 | 0.38 ± 1.04 |
| Anterior inferior (AI) | **0.45 ± 1.68** | **0.29 ± 1.27** |
| Posterior superior (PS) | **0.37 ± 1.14** | **0.31 ± 0.96** |
| Posterior inferior (PI) | **0.34 ± 1.34** | **0.03 ± 1.05** |
| Internal foot (IF) | **0.08 ± 1.37** | **0.02 ± 0.85** |
| External foot (EF) | 0.54 ± 1.70 | 0.57 ± 2.47 |

Notes.
Values pre and post presented in Mean ± Standard deviation.
Values in bold indicate a reduction in thermal asymmetry after treatment.

effectively detecting thermal asymmetries between sides of the body, in a simpler manner compared to currently used methods. This is in contrast to techniques such as magnetic resonance imaging or computed tomography, which, while accurate, are expensive and less accessible for frequent evaluations.

Moreover, thermography can be valuable for monitoring the progression of motor recovery and the response to therapeutic interventions. The relationship between sensorimotor function and thermography underscores the benefits of including it as an essential component in assessing patients' progress (*Hegedűs, 2018*). This is crucial because it allows treatment plans to be adjusted dynamically and evidence-based, potentially improving long-term functional outcomes.

However, certain limitations inherent in the use of thermography need to be addressed. The sensitivity of this technique can be affected by external factors such as ambient

**Table 5  Correlations between pre- and post-intervention thermal differences.**

|  | AS PRE | AS POST | AI PRE | AI POST | PS PRE | PS POST | PI PRE | PI POST | IF PRE | IF POST | EF PRE | EF POST |
|---|---|---|---|---|---|---|---|---|---|---|---|---|
| AS PRE | 1.00 | | | | | | | | | | | |
| AS POST | .56 | 1.00 | | | | | | | | | | |
| AI PRE | .18 | .58 | 1.00 | | | | | | | | | |
| AI POST | .19 | .73$^*$ | .89$^{**}$ | 1.00 | | | | | | | | |
| PS PRE | .35 | .87$^{**}$ | .78$^{**}$ | .87$^{**}$ | 1.00 | | | | | | | |
| PS POST | .53 | .95$^{**}$ | .43 | .60 | .84$^{**}$ | 1.00 | | | | | | |
| PI PRE | .36 | .71$^*$ | .61 | .59 | .77$^{**}$ | .72$^*$ | 1.00 | | | | | |
| PI POST | .19 | .69$^*$ | .55 | .50 | .75$^*$ | .71$^*$ | .81$^{**}$ | 1.00 | | | | |
| IF PRE | .37 | .76$^*$ | .75$^*$ | .70$^*$ | .81$^{**}$ | .73$^*$ | .92$^{**}$ | .82$^{**}$ | 1.00 | | | |
| IF POST | .28 | .70$^*$ | .74$^*$ | .75$^*$ | .77$^{**}$ | .66$^*$ | .89$^{**}$ | .77$^{**}$ | .95$^{**}$ | 1.00 | | |
| EF PRE | .32 | .72$^*$ | .87$^{**}$ | .79$^{**}$ | .82$^{**}$ | .58 | .65$^*$ | .70$^*$ | .67$^*$ | .64$^*$ | 1.00 | |
| EF POST | .09 | .39 | .42 | .30 | .46 | .52 | .39 | .53 | .56 | .44 | .29 | 1.00 |

**Notes.**

AS, Anterior superior; AI, Anterior inferior; PS, Posterior superior; PI, Posterior inferior; IF, Internal foot; EF, External foot. Spearman's rank correlation coefficient was applied.

$^*p < 0.05$.
$^{**}p < 0.01$.

temperature or image processing, which can introduce variability in the results (*Vázquez Cid de León et al., 2021*). For this reason, it is necessary to apply stringent protocols that control all variables that can influence this process. In our study, these protocols were implemented. However, despite our efforts to control all relevant factors, we observed variability in the environmental humidity. Therefore, we recommend implementing stricter measures to control this variable in future studies.

Our study results demonstrated that thermography can detect thermal differences between both sides of the body. We found statistically significant variances in most regions, which are considered clinically relevant. Some authors have stated that a 0.5 °C difference between the plegic and less affected sides is crucial in hemiplegia (*Hegedűs, 2018*). In our study, the average differences in values of both sides of the body were very close to the cut-off point and even exceeded it in the foot measurements of our patients. Considering this aspect, we suggest that assessing foot sole thermography may be crucial in determining the functional status of stroke patients. These results are also supported by the positive correlation established between the external aspect of the foot and the rest of the regions evaluated. As this correlation exists, we can affirm that the values found will show an appropriate functional state of the patient. However, this relationship was not shown with thermography of the thigh. In future studies, it would be interesting to check other points of the most proximal part of the lower extremity to determine the reason for these differences.

Strength training is increasingly important for treating neurological conditions like stroke. This training involves using the body's resistance, weights, dumbbells, machines, or resistance bands, requiring patients to make voluntary efforts to overcome the resistance. The training can be adjusted with different intensities, weights, repetitions, sets, and speeds. Numerous studies have shown that this training improves muscle tone and benefits

walking, cardiovascular endurance, energy consumption, and balance for patients (*Han et al., 2017*; *Wist, Clivaz & Sattelmayer, 2016*; *Lloyd et al., 2018*).

The scientific literature contains several studies on isolated strength and resistance training (*Flansbjer, Lexell & Brogårdh, 2012*). Still, most studies include mixed interventions combining strength training with other strategies, such as aerobic or balance training (*Rose et al., 2017*). For this study, we considered those studies that performed an exclusive lower extremity muscle strength protocol (*Büyükvural Şen et al., 2015*; *Lee et al., 2013*; *Lee & Kang, 2013*).

The studies involved muscle training programs lasting 5 to 6 weeks, which produced positive results in strength. These articles were chosen because no other studies have assessed the impact of similar programs within the same training session. Additionally, none of the studies evaluated thermal aspects, so selecting one with the most similar conditions was impossible.

After the intervention, no statistically significant differences in thermal variations were found. However, a change was observed that brought these variations closer to the ideal value of 0, which is the goal of this program.

## LIMITATIONS

This pilot study employed infrared thermography to evaluate the impact of a single-session strength training program on stroke patients. Although no statistically significant findings were observed after treatment, this suggests that the intervention period was too brief to produce measurable physiological effects. Nevertheless, thermography demonstrated sensitivity to thermal changes, indicating its potential for detecting subtle physiological variations. A more extended intervention period, involving multiple training sessions, may be necessary to measure the effects on circulation and thermoregulatory balance.

Previous studies (*Nowak, Mraz & Mraz, 2019*; *Hegedűs, 2018*) that implemented lower-limb physical therapy programs consistently reported positive outcomes, albeit with limited sample sizes. Some were restricted to individual case studies (*Alfieri et al., 2019*). In contrast, our study involved a comparatively larger sample; however, it was still underpowered to detect statistically significant differences after treatment. Therefore, future studies should include a larger sample size to improve the statistical power and generalizability of the results.

Moreover, we acknowledge that local genetic and epigenetic factors may influence thermal regulation and could partially explain inter-individual variability in the response to physical therapy. Additionally, heterogeneity in participants' body mass index (BMI) might have confounded thermal measurements. Future investigations should consider stratifying or limiting the sample based on BMI to reduce this source of variability.

Finally, while this study focused on thermographic analysis, we recommend that future research integrate functional outcome measures, such as motor and sensory assessments, to explore the relationship between thermal changes and recovery trajectories comprehensively.

## CONCLUSION

This article highlights the additional benefit of the digital processing of these images in diagnosing and monitoring patients with stroke. This study shows the effectiveness of thermography in detecting differences in body temperature following a stroke. We could accurately measure thermal variations in specific areas of interest using specialized tools like ThermImageJ to process and analyze detailed thermographic images. This method enabled us to identify subtle changes in skin temperature that may be linked to vascular and metabolic changes caused by a stroke.

Furthermore, research has demonstrated that a strength training program focused on the muscles in the lower extremities can decrease temperature differences in stroke patients following a single session. These findings indicate that specific physical therapy interventions not only enhance muscle function but also have the potential to positively impact other factors, such as peripheral circulation, as evidenced by improvements in thermal regulation.

Despite these results, more studies involving a larger population and longer intervention time are needed to obtain results that can be applied more broadly. However, this study emphasizes the benefits of integrating physical therapy with thermographic assessment and utilizing image analysis algorithms in clinical practice for stroke patients.

For future studies, it is proposed to carry out a longitudinal study, consider local genetic and epigenetic factors involved, and extend the duration of interventions to evaluate the long-term effects of muscle strength training on body temperature and functional recovery. Additionally, integrating advanced image processing techniques with machine learning could enhance the detection and quantification of thermal asymmetries. It would also be valuable to explore the correlation between thermal changes and other clinical parameters, such as balance or gait functionality.

### Funding
The authors received no funding for this work.

### Competing Interests
The authors declare there are no competing interests.

### Author Contributions

- Luis Augusto Silva Zendron conceived and designed the experiments, performed the experiments, analyzed the data, prepared figures and/or tables, authored or reviewed drafts of the article, and approved the final draft.
- Marta Gómez Mateos performed the experiments, authored or reviewed drafts of the article, and approved the final draft.
- Beatriz María Bermejo Gil conceived and designed the experiments, performed the experiments, analyzed the data, authored or reviewed drafts of the article, and approved the final draft.

- Andrea Calleja Caballero performed the experiments, authored or reviewed drafts of the article, and approved the final draft.
- Vanesa Santos Rodríguez performed the experiments, authored or reviewed drafts of the article, and approved the final draft.
- Fátima Pérez-Robledo conceived and designed the experiments, performed the experiments, analyzed the data, prepared figures and/or tables, authored or reviewed drafts of the article, and approved the final draft.
- Ana María Martín Nogueras conceived and designed the experiments, analyzed the data, authored or reviewed drafts of the article, and approved the final draft.

## Human Ethics

The following information was supplied relating to ethical approvals (*i.e.*, approving body and any reference numbers):

The Bioethics Committee of the University of Salamanca granted ethical approval for this study (Ref: 583).

## Data Availability

Raw data is available in the Supplemental Files.

## Supplemental Information

Supplemental information for this article can be found online at http://dx.doi.org/10.7717/peerj.19843#supplemental-information.

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
