# Peer review of "Thermography analysis as a tool for assessing thermal asymmetries and temperature changes after therapy in patients with stroke: a pilot study"

_PeerJ, doi:10.7717/peerj.19843_

## Round 0.1 · original submission · Major Revisions

· Academic Editor

Major Revisions

Please make major revisions in line with the reviewers' reports.

Reviewer 1 ·

Basic reporting

There is a lack of background information on the subject that needs to be reviewed by the authors. There are already several published works on this subject with larger samples than the one provided in this study, so the subject is neither original nor novel. In fact, several references are missing in the introduction (Line 43, 45,53-57 etc etc) that could better contextualize the problem and give a more global and profound vision of the current situation of this topic.
Examples.
Thermography Sensor to Assess Motor and Sensitive Neuromuscular Sequels of Brain Damage.
Thermography in the analysis of physiotherapeutic treatment in patients with brain damage: a systematic review.
The assessment of neuromuscular sequels post brain damage by thermography. A pilot study.

Experimental design

Methodology:
Participants
On what basis has the number of 10 patients been calculated? The sample seems too small to be able to state such concrete data. Such a sample might be more appropriate for a preliminary study.
Procedures:
Describe the provenance of the camera.
The anatomical references used to mark the ROIs need to be justified bibliographically. In addition, they must match the references shown in the images.

Validity of the findings

The sample is too heterogeneous based on BMI results. Because the fat percentage can influence the thermal variables the results could be subject to mythological bias due to the small sample size of the study.
Figure 3 shows the image if the reference ROI and the patient appears in shorts, which makes it difficult to understand how the data from the quadricipital region were collected. In addition, thermal references do not appear in the color scale, so the thermographic results are not interpretable.
The image appears out of focus and this can alter the thermal response obtained.
The comparison with results obtained by other more current authors and with larger samples is missing. The article in this sense does not provide anything new or any innovative advance.
The study does not provide any data other than temperature values, which in itself has no predictive value. It would be interesting to associate this value to some specific discriminant test of strength or to some specific assessment scale as proposed by other authors.
The machine used is not for human use. Describe this as a limitation of the study the fact that it is not listed as a medical device by the Food and Drug Administration (FDA) or the European Medicines Agency (EMA).

Additional comments

The idea of associating thermographic imaging to patients with neurological disorders is a topic in great development at present. However, there are already several studies on this subject that have not been cited and provide the same results with the same bibliography. Before publishing an article, it is necessary to carry out a thorough review of the current status of the subject, to avoid reproducing results already published, in order to produce information that contributes new knowledge. This review also serves to avoid reproducing methodological errors that may influence the results obtained, something that is observable in this work.

·

Basic reporting

The basic review:
This article is clearly written and organized, the introduction presents the topic in a very complete manner and the figures and titles are also comprehensive.

This paper follows the data required by the journal:
The article presents an abstract with content about the entire article in a clear manner and with details about the study.
The introduction of the article is complete with a comprehensive explanation on the subject reported in the theme.
The text presents the methodology, such as patient selection and the entire procedure of the experimental design. It also presents a figure reporting the entire procedure and the images taken of the volunteers' bodies evaluated.
The statistical analysis was done based on IBM statistics and the results are well described, showing the height and weight of the patients evaluated.
Finally, the article shows good quality images to understand how the patient's body was evaluated through thermography with the camera, with tables that demonstrate what really happened in this study.
The discussion is concise, explaining the entire understanding of this line of research. The article also contains a text about the limitations of this study and the conclusions.

Experimental design

As seen in the conclusion, this research needs to be evaluated with a larger group of volunteers in a longitudinal manner and to consider local and genetic epigenetic factors.
In my review, it was a little long and could be shortened, and part of this text could be taken to the end of the discussion.

Validity of the findings

I reiterate that the tables, figures and procedure diagram show very well the details applied and results evaluated in this work.

The use of the thermographic camera for the diagnosis of stroke This article highlights the additional benefit of digital processing of these images in diagnosis and monitoring of patients with stroke.

Additional comments

I consider the bibliographical references reasonable; this is a part of the work that can always be improved.

The conclusions are pertinent regarding the use of the thermographic camera for the diagnosis of stroke; my considerations made above should be described in this article for better conclusion and possible publication of this work.

---

## Round 0.2 · Major Revisions

· Academic Editor

Major Revisions

Thank you for your submission. After careful evaluation by the reviewers, it is clear that substantial revisions are required for the manuscript to meet the standards for publication. The reviewers have highlighted critical concerns regarding the understanding and application of thermographic methods, which are fundamental to the validity of the study. Specifically, it was noted that the use of thermography in your study appears to treat the device as a digital thermometer, which is not aligned with current standards in the field. Additionally, a more comprehensive review of the relevant literature is essential to contextualize and support your findings. Several key references in this area were not included or discussed, which raises concerns about the robustness of the discussion and interpretation of results. We strongly advise you to conduct a deep and careful revision of your manuscript

Reviewer 1 ·

Basic reporting

The bibliographical references need to be checked again. Repeated references appear.

The English is unclear, redundant and with important linguistic confusions. There are still basic concepts such as hemisphere or hemibody that are confused with each other (Line 256),

Figures remain unclear and poorly described

Experimental design

Image acquisition and analysis protocols lack adequate methodological quality.

The methods are described in an approximate and not very detailed way from a bibliographical point of view. The thermograph in this study is being used as a digital thermometer.

Validity of the findings

Results:
The results continue to appear disorganised, poorly structured and lacking in fundamentals. There are still basic concepts such as hemisphere or hemibody that are confused with each other (Line 256), showing the authors' lack of control over the subject matter. Numerous studies with which the results obtained could be compared have not been cited by the authors, thus reducing the quality of the discussion.

Additional comments

Introduction:
The introduction is confusing and in many cases redundant. The state of the art remains unclear. There is a lack of recent articles that could further enrich this section.

Material and method:
• Procedure:
1. Currently, the TISEM protocol is the most up to date and the most reliable in the recommendations to be followed for the development of a good thermographic methodology. Have the TISEM recommendations been followed in carrying out this study? Cite the reference study in the case these recommendations were used
2. ROIs remain unclear and in fact represent a major methodological error. They do not analyse points in thermographic domains, but whole anatomical regions. Within the same region there can be different temperatures that if analysed by points can generate errors in the analysis of the data.
3. The origin of the camera remains unexplained.

• Statistical analysis:
1. The statistical analysis is underdeveloped and the statistical tests applied are not clear.

Results:
The results continue to appear disorganised, poorly structured and lacking in fundamentals. There are still basic concepts such as hemisphere or hemibody that are confused with each other (Line 256), showing the authors' lack of control over the subject matter. Numerous studies with which the results obtained could be compared have not been cited by the authors, thus reducing the quality of the discussion.

Limitation:
Finally, while this study focused on thermographic analysis, we recommend that future research 304 integrate functional outcome measures-such as motor and sensory assessments- to comprehensively 305 explore the relationship between thermal changes and recovery trajectories. Such studies already exist, they have simply not been taken into account by the authors.

Reference:
There are several references repeated several times in the text. Check the entire bibliography.

·

Basic reporting

This article is clearly written and organized, the introduction presents the topic in a very
complete manner and the figures and titles are also comprehensive.

Experimental design

The article presents an abstract with content about the entire article in a clear manner and with
details about the study

Validity of the findings

The findings are interesting and consistent with the methodology presented in this article. Interesting results.

Additional comments

I recommend publishing this article as it appears, corrected in this latest version.

---

## Round 0.3 · accepted · Accept

· Academic Editor

Accept

I confirm that the authors have addressed all of the reviewers' comments in their revised manuscript. Based on the reviewers' feedback, the manuscript is ready for publication.

Reviewer 1 ·

Basic reporting

Ok

Experimental design

Ok

Validity of the findings

Ok

Additional comments

It`s Ok